# Comparison of Local Metabolic Changes in Diabetic Rodent Kidneys Using Mass Spectrometry Imaging

**DOI:** 10.3390/metabo13030324

**Published:** 2023-02-22

**Authors:** Xin Zhang, Yanhua Liu, Shu Yang, Xin Gao, Shuo Wang, Zhaoying Wang, Chen Zhang, Zhi Zhou, Yanhua Chen, Zhonghua Wang, Zeper Abliz

**Affiliations:** 1Key Laboratory of Mass Spectrometry Imaging and Metabolomics, Minzu University of China, National Ethnic Affairs Commission, Beijing 100081, China; 2Center for Imaging and Systems Biology, College of Life and Environmental Sciences, Minzu University of China, 27 Zhongguancun South Avenue, Beijing 100081, China; 3Key Laboratory of Ethnomedicine of Ministry of Education, School of Pharmacy, Minzu University of China, Beijing 100081, China

**Keywords:** spatially resolved metabolomics, mass spectrometry imaging, diabetic nephropathy, HFD/STZ-induced diabetic rats, db/db mice

## Abstract

Understanding the renal region-specific metabolic alteration in different animal models of diabetic nephropathy (DN) is critical for uncovering the underlying mechanisms and for developing effective treatments. In the present study, spatially resolved metabolomics based on air flow-assisted desorption electrospray ionization mass spectrometry imaging (AFADESI-MSI) was used to compare the local metabolic changes in the kidneys of HFD/STZ-induced diabetic rats and db/db mice. As a result, a total of 67 and 59 discriminating metabolites were identified and visualized in the kidneys of the HFD/STZ-induced diabetic rats and db/db mice, respectively. The result showed that there were significant region-specific changes in the glycolysis, TCA cycle, lipid metabolism, carnitine metabolism, choline metabolism, and purine metabolism in both DN models. However, the regional levels of the ten metabolites, including glucose, AMP, eicosenoic acid, eicosapentaenoic acid, Phosphatidylserine (36:1), Phosphatidylserine (36:4), Phosphatidylethanolamine (34:1), Phosphatidylethanolamine (36:4), Phosphatidylcholine (34:2), Phosphatidylinositol (38:5) were changed in reversed directions, indicating significant differences in the local metabolic phenotypes of these two commonly used DN animal models. This study provides comprehensive and in-depth analysis of the differences in the tissue and molecular pathological features in diabetic kidney injury in HFD/STZ-induced diabetic rats and db/db mice.

## 1. Introduction

Diabetes mellitus (DM) is a chronic metabolic disease with a growing global incidence [1,2,3,4,5]. It is estimated that the number of diabetic patients will increase to 700 million by 2045 [6], with an estimated incidence rate of 10% in China [7]. Type 2 diabetes mellitus (T2DM) is the most prevalent form of DM, affecting over 90% of all diagnosed patients. It is estimated that 30–40% of diabetic patients will develop diabetic nephropathy (DN), which is one of the major complications of DM and the leading cause of end-stage renal disease (ESRD) [8]. Despite its widespread impact, the etiology and pathogenesis of DN is still not fully understood, and the current treatments are limited.

Animal models have been crucial in uncovering the pathophysiology of diseases, in identifying new targets for treatment, and in testing potential therapeutic agents. Numerous spontaneous and experimentally induced animal models of DN have been established for various research purposes [9]. Diabetic db/db mice are a good representation of the early changes in human DN, including albuminuria, podocyte loss, and mesangial matrix expansion, but do not show the later, more advanced morphological changes [10]. The HFD/STZ-induced DN model, which involves a high-fat diet feeding and a low-dose streptozotocin injection, is a commonly used model and exhibits elevated albuminuria, increased kidney index, and glomerular hypertrophy and mesangial matrix accumulation. However, the similarity of this model to human DN has been questioned [11,12]. As none of the current animal models can perfectly replicate all the pathological features of human DN, the selection of animal models for DN research must be conducted with caution. Understanding the phenotypic features of different animal models of DN can aid in the effective and targeted use of these models and enhance our understanding of DN.

Metabolomics is an important molecular profiling technology that can comprehensively capture the biological consequences of disease by identifying the metabolic biomarkers that correlate with disease phenotypes [13,14]. Several researchers have identified a large number of metabolic biomarkers related to DN using nuclear magnetic resonance (NMR), gas chromatography–mass spectrometry (GC–MS), and liquid chromatography–mass spectrometry (LC–MS)-based metabolomic approaches. For example, Salek et al. [15] investigated the metabolic similarities between mice, rats, and humans with T2DM using 1H-NMR-based metabolomics and identified some metabolic perturbations common to all three species in urine samples. These metabolomics studies have greatly advanced our understanding of the phenotypes of DN, but the local metabolic changes in the diabetic kidney can only be partially evaluated through the analysis of homogeneous samples, such as urine, serum, and tissue homogenates.

Mass spectrometry imaging (MSI) is a highly advanced molecular imaging technology that allows the spatially resolved metabolic profiling of a variety of functional metabolites in tissue sections, which is important for gaining a more accurate understanding of the tissue-specific molecular pathological features underlying diseases [16]. A variety of MSI techniques, such as matrix-assisted laser desorption ionization (MALDI)-MSI, desorption electrospray ionization (DESI)-MSI, and secondary ion mass spectrometry (SIMS), have been developed and frequently used in the biomedical field. For example, Satoshi Miyamoto et al. [17] revealed an increase in the glomerular ATP/AMP ratio in the diabetic kidney using MALDI-MSI and identified sphingomyelin (d18:1/16:0) as the key regulator for ATP production in mesangial cells. Additionally, our research group developed a spatially resolved metabolomic approach based on air flow-assisted desorption electrospray ionization (AFADESI)-MSI and MALDI-MSI and discovered the tissue-specific metabolic reprogramming processes in the kidney of a rat model of DN 12 weeks after the induction of diabetes by HFD/STZ [18].

In this study, we described the application of AFADESI-MSI-based spatially resolved metabolomics to investigate the region-specific metabolic changes in a rat model of DN produced 20 weeks after the induction of diabetes by HFD/STZ and a spontaneous DN model of 28-week-old db/db mice. Local metabolic changes in the kidneys of the two animal models were compared with an aim to determine the key discriminating metabolites that were common or different to the two species. The research workflow is summarized in Figure 1.

## 2. Materials and Methods

### 2.1. Chemicals and Reagents

HPLC-grade acetonitrile (ACN) and methanol (MeOH) were purchased from Merck (Muskegon, MI, USA). Purified water was obtained from Wahaha (Hangzhou, China). STZ and citrate were purchased from Sigma-Aldrich (St. Louis, MO, USA).

### 2.2. Animal Models

DN model of HFD/STZ-induced rat: Six-week-old male Wistar rats weighing 175–210 g were purchased from Vital River Laboratory Animal Technology Co., Ltd. (Beijing, China); they were housed in a constant environment maintained at 23 ± 3 °C with a 12 h day/night cycle. The rats were randomly divided into control (n = 6) and DN model (n = 6) groups. A normal pellet diet and an HFD (5.24 Kcal/g, 60% fat; HFD12492, high-fat feed, Beijing HFK Bioscience Co., Ltd, Beijing, China) were given to the rats in the control and model groups, respectively. After 4 weeks, the rats in the model group were induced to develop insulin resistance (IR) and then injected intraperitoneally (ip) with STZ at a dose of 35 mg/kg to induce hyperglycemia, while the rats in the control group were injected ip with citrate buffer (PH = 4.4, vehicle). The fasting blood glucose (FBG) level was measured 5–7 days after the STZ or buffer injection. An FBG level of ≥16.7 mmol/L for three consecutive days was used as the standard for establishing the model. After 20 weeks of feeding, 24 h urine was collected from each rat using metabolic cages. Blood was collected from the abdominal artery. The kidneys were collected and snap-frozen in liquid nitrogen.

DN model of db/db mouse: Four-week-old male BKS.DB mice (6 db/db homozygotes and 6 db/m littermate controls) weighing 16.5–35.6 g were purchased from gempharmatech Co., Ltd. (Jiangsu, China); they were housed at a constant temperature of 23 ± 3 °C with a 12 h dark/12 h light cycle. The db/m littermate controls were set as the control group, and the db/db homozygotes were set as the diabetes model group. After 24 weeks of feeding, the urine, blood, and kidney samples were collected in the same way as those used in the HFD/STZ-induced rats.

All the samples were stored at −80 °C until analysis. The animal experiments were approved by the animal welfare ethics committee of Beijing Union-Genius Pharmaceutical Technology Development Co., Ltd. (Beijing, China) and conducted in accordance with the NIH Guide for the Care and Use of Laboratory Animals.

### 2.3. Biochemical and Histopathological Analysis

The FBG was determined using a blood glucose meter (Roche, Basel, Switzerland). Glycosylated hemoglobin (HbA1c) was detected with a Quo-Test HbA1c Analyzer (QUOTIENT Diagnostics Ltd. Walton-on-Thames, Surrey, UK). The urine creatinine and urea nitrogen were measured using an AU480 automatic chemistry analyzer (Beckman Coulter Inc., Brea, CA, USA).

The frozen kidney tissues were cut into 8 μm serial sections at −22 °C using a Leica CM1860 cryostat (Leica Microsystems Ltd., Wetzlar, Germany) and mounted onto adhesion microscope slides (Thermo Scientific, CA, USA). Hematoxylin and eosin (H&E) staining of kidney sections was performed to reveal the histopathological lesions.

### 2.4. AFADESI−MSI Analysis

The MSI experiments were performed on an AFADESI-MSI platform, which consisted of a home-built AFADESI ion source and a Q-OT-qIT hybrid mass spectrometer (Orbitrap Fusion Lumos; Thermo Fisher Scientific, San Jose, CA, USA) [19,20]. Mass spectra were obtained in positive and negative full MS modes with a scan range of 100–1000 Da at a mass resolution of 120,000. Additional details on the experimental parameters can be found in the Appendix A.

### 2.5. Data Processing and Analysis

Raw files obtained from the analysis in the positive and negative AFADESI-MSI ion modes were converted to .cdf format by Xcalibur 4.0.2 (Thermo Scientific, San Jose, CA USA). The converted file was then imported into MassImager (MassImager 2.0, Beijing, China) for the background subtracting, normalizing, and reconstructing of the ion image. Regions of interest (ROIs) were selected by matching the H&E stain image to generate individual .txt format data. The datasets were imported into Markerview™ software 1.2.1 (AB SCIEX Toronto, ON, Canada) for peak picking and alignment. Relative intensities were calculated for each ROI using total ion current (TIC) normalization, and further compared using Student’s *t*-tests to find the differentiating metabolites associated with DN (*p* < 0.05).

### 2.6. LC-MS/MS Analysis of Kidney Homogenates

The kidney homogenates were analyzed by high-resolution LC-MS/MS to obtain structural information. Detailed procedures for the LC-MS/MS experiments are provided in the Appendix A.

### 2.7. Venn Diagram and Pathway Enrichment Analysis of Discriminating Metabolites

Venn diagrams were drawn using Venn 2.1 (https://bioinfogp.cnb.csic.es/tools/venny/index.html, accessed on 5 April 2022), and then, the theoretical *m/z* of the discriminating metabolites involved in each group was entered into the corresponding list to automatically refresh the presented results.

Pathway analysis was performed using the Pathway Analysis module in MetaboAnalyst 3.0 (https://www.metaboanalyst.ca, accessed on 7 April 2022). The HMDB IDs of the discriminatory metabolites were entered into the compound list, and the species was selected as rat. An enrichment analysis of the pathways was then conducted. The results of the “Match Status” were saved as a .csv file, and pathway enrichment analysis graphs were generated using the R version 3.6.1 programming language.

## 3. Results

### 3.1. Assessment of Renal Injury in HFD/STZ-Induced Diabetic Rats and db/db Mice

The results for the body weight, water consumption, food intake, and biochemical parameters of the rats in the control and the HFD/STZ-induced DN model group are summarized in Appendix A. As compared to the control group, the body weight of the rats in the model group significantly decreased (*p* < 0.01) (Figure 2A), The urea nitrogen was slightly increased (Figure 3C), while the level of urine creatinine was significantly decreased (*p* < 0.01) (Figure 3D), The kidney weight and the kidney/body weight ratio in the HFD/STZ-induced DN group were remarkably higher than those in the control group (*p* < 0.05) (Figure 3E,F).

Representative photomicrographs of the H&E-stained kidney tissues of the rats in the control and HFD/STZ-induced DN model group are shown in the Appendix A. The histological examinations of the kidneys revealed swelling, mild cortical tubular epithelial space-like changes, and protein flocculation in the tubular lumen in the DN group. These biological and pathological changes indicate that diabetes causes mild pathological changes in the chronic progressive nephropathy in rats. The results for the body weight, water consumption, food intake, and biochemical parameters of the db/m and db/db mice are shown in Appendix A. The change trends of the water consumption, FBG, HbA1c, urea nitrogen, urine creatinine, and kidney weight in the db/db mice were similar to those observed in the HFD/STZ-induced diabetic rats. However, the change trends of the body weight and kidney/body weight ratios were reversed in the db/db mice as compared to the HFD/STZ-induced diabetic rats, which may be due to the reversed food intake changes in the two diabetic models.

The results of the H&E staining of the kidney tissues in the db/db mice were also examined and showed similar signs of renal hypertrophy, morphological changes, and vacuolar degeneration of the renal tubular epithelial cells, as observed in the HFD/STZ-induced diabetic rats. However, a more significant sign of glomerular damage, represented by the sieve-like dilatation of the glomerular capillaries, was observed in the kidneys of the db/db mice in DN model of the db/db mice compared to the HFD/STZ-induced DN rats.

### 3.2. AFADESI-MSI Analysis of Kidneys of HFD/STZ-Induced DN Rats and db/db DN Mice

The frozen renal sections of the control and model groups were profiled by AFADESI-MSI to investigate the spatially resolved metabolic alteration in the HFD/STZ-induced DN rats and the db/db DN mice. The renal section was roughly divided into four regions, i.e., whole kidney (W), cortex (C), outer medulla (OM) and inner medulla (IM), based on H&E staining. In the positive AFADESI-MSI mode, 1750, 1642, 1569, and 1846 ion features were detected in the W, C, OM, and IM regions in the HFD/STZ-induced DN rats, respectively, while 1481, 1328, 1442, and 1691 ion features were detected in the W, C, OM, and IM regions in the db/db DN mice, respectively. In the negative AFADESI-MSI mode, 554, 528, 493, and 515 versus 580, 572, 529, and 585 peaks in the W, C, OM, and IM regions from the HFD/STZ-induced DN rats versus the db/db DN mice, respectively. A Student’s *t*-test was performed to compare the metabolic profiles of each region of the kidneys collected from the control and model groups. Based on a threshold of a *p* value < 0.05, a total of 274, 291, 318, and 224 differentiating variables were selected in the W, C, OM, and IM regions of the HFD/STZ-induced DN rats, and 299, 233, 326, and 294 differentiating variables were selected in the corresponding regions of the db/db DN mice, respectively.

The structures of the differentiating metabolites were provisionally identified based on our previously described protocols [18]. Briefly, the possible element composition of the ion was calculated using Xcalibur 4.0 (Thermo Fisher Scientific), based on the exact mass and isotope pattern, and it was then searched in HMDB (http://hmdb.ca/, accessed on 20 April 2022) and METLIN (http://metlin.scripps.edu/, accessed on 21 April 2022) to find the possible metabolites. Finally, high-resolution on-tissue AFADESI−MS/MS analysis of the kidney sections and LC-MS/MS analysis of the kidney homogenates were performed to confirm the structures of the metabolites by interpretating their fragmentation characteristics. As result, 12 and 55 discriminating metabolites were finally identified in the HFD/STZ-induced DN rats in the positive and negative ion modes, respectively. 17 and 42 metabolites were identified in db/db DN mice in the positive and negative ion modes, respectively. The results of the metabolite identifications, including the fold changes (FC) of the differentiating metabolites, are presented in Appendix A and Appendix A.

As depicted in the Venn diagram (Figure 4), the majority of the differentiating metabolites have significant changes in the entire kidney. However, certain metabolites showed changes only in specific regions. In the HFD/STZ-induced diabetic rats, 8, 10, and 1 distinctive discriminating metabolites were detected in the cortex, outer medulla, and inner medulla, respectively. In the db/db mouse model, 4, 13, and 5 unique discriminating metabolites were detected in the cortex, outer medulla, and inner medulla, respectively.

The fan charts (Figure 4C,D) depict the proportion of differential metabolites found in the different regions of the kidney. The analysis revealed that the ranking of the number of discriminating metabolites identified in the different regions of the HFD/STZ-induced rat kidneys was OM/C > W > IM, while in the db/db mice, the ranking was OM > W > C > IM, indicating that the outer medulla region may be more vulnerable to diabetic renal damage in comparison to the other regions [21,22].

### 3.3. Pathway Enrichment Analysis

The pathway enrichment analysis of the discriminating metabolites was performed using MetaboAnalyst 3.0 (https://www.metaboanalyst.ca, accessed on 10 May 2022) to identify the metabolic pathways involved in the HFD/STZ-induced diabetic rats and db/db mice. As shown in Figure 5, a total of 17 altered metabolic pathways were identified in the HFD/STZ-induced diabetic rats, with 12 items having impact values greater than 0. The top five affected pathways were linoleic acid metabolism, taurine and hypotaurine metabolism, α-linolenic acid metabolism, glycerophospholipid metabolism, and the TCA cycle. Similarly, a total of 19 metabolic pathways were associated with the db/db mice. As shown in the Figure 5, there are nine items having impact values greater than 0. The top five affected pathways in db/db were glycerophospholipid metabolism, arachidonic acid metabolism, purine metabolism, the TCA cycle, and glycine, serine, and threonine metabolism.

## 4. Discussion

The findings suggest that there are similarities and differences in the metabolic alteration in kidneys of HFD/STZ-induced DN rats and db/db DN mice. A comparison of the discriminatory metabolites associated with DN between the two models is shown in Appendix A. Only 15 (14.6%) discriminatory metabolites showed the same trend of change, with 7 metabolites being down-regulated and 8 being up-regulated in both DN models. Forty-four (42.7%) discriminatory metabolites were found to be specifically altered in the HFD/STZ-induced rats, while thirty-six (35%) discriminatory metabolites were uniquely changed in the db/db mice. The classes of the discriminatory metabolites, which had common or specific variance in the two models, are summarized in Appendix A, with most of the discriminating metabolites being lipids.

The comparison of the metabolic alterations associated with DN in the two models is shown in Appendix A. The results indicated that both the HFD/STZ-induced DN rats and the db/db DN mice had alterations in their metabolic pathways, including glycerophospholipid metabolism, purine metabolism, the TCA cycle, and glycosylphosphatidylinositol (GPI)-anchored biosynthesis metabolism. However, the findings suggest that linoleic acid metabolism was more affected by the HFD/STZ-induced DN, while the changes in the glycerophospholipid metabolism and the arachidonic acid metabolism were more pronounced in the db/db-induced DN.

### 4.1. Alteration of Lipid Metabolism in HFD/STZ-Induced Diabetic Rats and db/db Mice

A total of 49 and 41 discriminating lipids were identified in the HFD/STZ-induced diabetic rats and db/db mice, respectively. The fan chart of the discriminating lipid species is provided in Appendix A. The spatial distribution and changes of these lipids in the HFD/STZ-induced diabetic rats and db/db mice are summarized in Figure 6 and Figure 7, respectively. These metabolites showed unique distributions across the kidney sections, indicating extensive alteration in the lipid metabolism occurred in a region-specific manner in both models of DN.

#### 4.1.1. Fatty Acid Metabolism

In the HFD/STZ-induced DN rats, three monounsaturated fatty acids (MUFAs), including oleic acid, eicosenoic acid, and nervonic acid were up-regulated, and four polyunsaturated fatty acids (PUFAs), including linolenic acid, linoleic acid, eicosapentaenoic acid, and docosahexaenoic acid were down-regulated. Additionally, the fatty acid esters of the hydroxy fatty acids FAHFA (34:1) were decreased, whereas the FAHFA (34:0) was increased. In the db/db DN mice, similar change trends were observed for palmitelaidic acid, eicosenoic acid, linolenic acid, and arachidonic acid compared to the HFD/STZ-induced diabetic rats. However, the change trends were found to be the opposite for eicosenoic acid and eicosapentaenoic acid.

Fatty acids (FAs) are known to play a role in a wide range of diseases, including T2DM, inflammatory diseases, and cancer [23]. Linolenic acid has been shown to reduce inflammation by inhibiting the accumulation of the extracellular matrix (ECM) in DN [24]. On the other hand, oleic acid and palmitelaidic acid are the most abundant dietary and plasma fatty acids and have different effects on insulin resistance [25]. Oleic acid has been linked to the production of reactive oxygen species (ROS), leading to oxidative damage in proximal renal tubular cells [26]. In contrast, palmitelaidic acid has been shown to regulate glucose homeostasis, lipid metabolism, and cytokine production, thereby improving metabolic disorders [27]. A new family of synthetic fatty acid esters of hydroxy fatty acids (FAHFAs) secreted by adipose tissue has been found to have beneficial metabolic effects, such as enhanced insulin-stimulated glucose transport and glucose-stimulated GLP1 and insulin secretion, and to exhibit anti-inflammatory effects [28,29,30]. These FAHFAs are considered to be promising endogenous lipids with therapeutic potential for T2DM.

#### 4.1.2. Phospholipid Metabolism

Significant changes were observed in the levels of 5 phosphatidylcholine (PC), 8 phosphatidylethanolamine (PE), 5 phosphatidylserine (PS), 5 phosphatidylinositol (PI), 1 phosphatidic acid (PA), 9 phosphatidylglycerol (PG), 4 lysophospholipids, and 4 sphingomyelin (SM) metabolites in the HFD/STZ-induced DN rats. PC(34:1), PC(36:1), PC(36:2), PC(36:4), PE(p-18:0/20:4(5Z,8Z,11Z, 14Z)), PG(36:2), PG(36:3), PG(40:6), LysoPG(18:1), and SM(d18:1/16:0) were found to be up-regulated, while PC(34:2), PE(34:1), PE(36:2), PE(36:4), PE(38:4), PE(38:6), PE(p-38:6), phosphorylethanolamine, LysoPE(16:0), LysoPE(20:4), PS(34:1), PS(36:1), PS(36:2), PS(36:4), PS(40:6), PG(32:0), PG(34:2), PG(36:4), PG(40:8), PG(42:10), PG(44:12), PA(34:2), PI(34:2), PI(18:1(9Z)/18:1(9Z)), PI(36:4), PI(38:5), PI(40:6), LysoPI(16:0), and SM(42:2) were down-regulated. In the db/db DN mice, 8 PC, 10 PE, 2 PS, 4 PG, 3 PA, 4 PI, 3 lysophospholipid, and 2 SM metabolites were altered significantly. PC(34:0), PC(34:1), PC(34:2), PC(36:2), PC(36:3), PC(36:4), PC(38:4), LysoPC(18:0), PE(34:1), PE(p-34:1), PE(36:4), PE(p-36:4), PE(p-38:4), PE(40:1), PE(42:1), PS(36:1), PS(36:4), PG(34:1), LysoPG(18:1), LysoPG(18:2), PA(16:0/18:1(9Z)), PA(38:4), PI(34:0), PI(38:4), PI(38:5), SM(d18:1/16:0), and SM(d18:1/24:1(15Z)) were found to be up-regulated, while PC(p-38:5), PE(36:6), PE(p-40:4), PE(p-40:7), PG(32:0), PG(38:4), PG(40:8), PA(41:5), and PI(40:6) were down-regulated.

PC and PE are the most abundant phospholipids in mammalian cell membranes. The PC/PE ratios in various tissues are associated with atherosclerosis, insulin resistance, and obesity [31]. In the present study, we calculated the PC/PE ratios in different regions of the kidneys in both models. The results showed that the PC/PE ratios were elevated in the W, C, OM and IM regions of the HFD/STZ-induced DN rats as compared to the control group. In the db/db mice, the PC/PE ratio was elevated in the W, OM, and IM regions, but decreased in the C region in the model group as compared to the control group. These changes in the PC/PE ratio may alter the fluidity and permeability of the cell membrane, leading to cellular damage [32,33].

The PS metabolites were found to have opposite change trends in the two DN models. In the cortex and outer medulla of the HFD/STZ-induced rats, most of the PS metabolites decreased, whereas they accumulated in the db/db mice (as shown in Figure 7E and Appendix A). The decrease in PS metabolites may be a result of increased glycation in the diabetic state. Under diabetes, PS can be glycosylized to the form Amadori compounds and advanced glycation end products (AGEs), which are known to accumulate in diabetic patients [34].

The changes in the PG metabolites were consistent between the two DN model groups, with most of the PG metabolites showing a decrease in the model group and LysoPG being generally elevated in the cortical and outer medullary regions in both model groups (Figure 6F,G, Figure 7E,F, Appendix A). Meanwhile, PA was found to be decreased in the HFD/STZ-induced rats (Figure 6G) but elevated in the db/db mice (Figure 7F and Appendix A). These alterations in PA levels may lead to the pathological remodeling of cardiolipin (CL) and mitochondrial dysfunction in diabetes [18].

The trends in PI metabolites differed between the two DN models. The PI levels decreased in the cortical and outer medulla of the HFD/STZ-induced rats (Figure 6G,H and Appendix A), while they showed an upward trend in the db/db mice (Figure 7G and Appendix A). A decrease in PI biomarkers is commonly observed in patients with T2DM and DN, which may be linked to activation of the sorbitol pathway (SP). These findings were consistent with the previous studies on HFD/STZ-induced rats [35].

SM plays critical structural and functional roles in cells and has been linked to podocyte injury and atherosclerotic plaque inflammation [36,37,38]. The results of this study showed an accumulation of SM(d18:1/16:0) in the kidneys of both the DN models (Figure 6H, Figure 7G, Appendix A). Increased levels of SM (d18:1/16:0) may suppress the activity of AMP-activated protein kinase (AMPK) and decrease PGC-1α protein expression, leading to an increase in glycolytic pathway activity [39].

### 4.2. Alteration of Glycolysis and the TCA Cycle in HFD/STZ-Induced Diabetic Rats and db/db Mice

The trends and spatial distributions of the metabolites involved in the glycolysis and TCA cycle pathways are illustrated in Figure 8 and Appendix A. Increased glucose levels were observed in the renal cortex in both of the DN models. However, the metabolic fate of glucose varied between the two models, with a slight decrease observed in the outer medulla in the HFD/STZ-induced DN rats, but a significant increase was seen in the outer medulla and inner medulla regions in the db/db DN mice. This difference may be attributed to differences in the expression of the sodium-glucose cotransporters (SGLTs), which are predominantly distributed in the outer medulla and play a crucial role in the resorption and utilization of glucose in the kidney. The lower expression of SGLT2 in the outer medulla in the HFD/STZ-induced DN rats and the increased expression of SGLT2 in the outer medulla in the db/db mice may have contributed to the observed trends [40].

The trends of the metabolites associated with the glycolysis and TCA cycle pathways showed differences between the two DN models. Citrate, glutamate, and aspartate were found to decrease in the HFD/STZ-induced DN rats, while malate and glutamate showed a decrease in the db/db DN mice. On the other hand, succinate was significantly increased in the cortex and outer medulla in the db/db DN mice. These results were consistent with the previous findings [18]. The alteration of these TCA intermediates indicates a reduction in the succinate dehydrogenase (SDH) activity and a disruption of the mitochondrial homeostasis and function in the kidneys; these have been associated with the progression of DN [41,42].

### 4.3. Alteration of Purine Metabolism in HFD/STZ-Induced Diabetic Rats and db/db Mice

The change trends and spatial distributions of the metabolites related to the purine metabolism in the two model groups are depicted in Figure 9 and Appendix A. Adenosine monophosphate (AMP) showed a slight decrease in the outer medulla in the HFD/STZ-induced DN rats, but it was significantly increased in all the renal regions in the db/db DN mice. AMP acts as the primary regulator of AMPK, a crucial player in the regulation of metabolism and energy balance [43]. The activity of AMPK was reportedly decreased in both STZ-induced diabetic rats and db/db mice [44]. These opposing change trends in the AMP levels suggest that the renal energy metabolism may be different in the two DN models.

### 4.4. Alteration of Carnitine Metabolism in HFD/STZ-Induced Diabetic Rats and db/db Mice

The change trends and spatial distributions of the metabolites associated with carnitine metabolism in the two model groups are displayed in Figure 10 and Appendix A. A significant reduction in L-carnitine levels was observed in the HFD/STZ-induced DN rats. Both DN models showed an accumulation of long-chain acylcarnitines, with increased levels of stearoylcarnitine in the HFD/STZ-induced DN rats and elevated concentrations of palmitoylcarnitine, linoleylcarnitine, and octadecenoylcarnitine in the db/db DN mice. Carnitines play an important role in several intermediate metabolic processes, especially in oxidative lipid metabolism, where they facilitate the transportation of long-chain fatty acids through the mitochondrial membrane to promote the β-oxidation of fatty acids [45]. L-Carnitine has been demonstrated to have anti-inflammatory and antioxidant properties, as well as to improve insulin sensitivity and dyslipidemia [46,47]. Acylcarnitines are intermediate metabolites of fatty acid oxidation, and their accumulation has been associated with insulin resistance in diabetes [48,49]. These altered levels of carnitine metabolites indicate a disruption in the mitochondrial oxidative lipid metabolism in the kidneys in both DN models.

### 4.5. Alteration of Choline Metabolism in HFD/STZ-Induced Diabetic Rats and db/db Mice

Decreased levels of choline were observed in both the DN models, while increased levels of betaine were observed in the db/db mice (Figure 10). Choline is crucial in the synthesis of PC, PE, and SM and plays a key role in the regulation of cell membrane fluidity and osmoregulation. As an oxidized metabolite of choline, betaine is an important osmoprotectant in the kidney and contributes to the formation of the axial osmolality. The altered levels of choline and betaine indicated that the axial osmolality of the kidney may be impacted in DN.

The changes in the metabolites and metabolic pathways associated with DN are summarized in Figure 11. Our findings showed that although both the HFD/STZ-induced diabetic rats and the db/db mice exhibit dysregulations of glycolysis, the TCA cycle, lipid metabolism, carnitine metabolism, choline metabolism, and purine metabolism, the specific changes in the levels of these metabolites and their regional distribution vary between the two DN models. This was highlighted by the opposite glucose and AMP alteration in the medulla. Additionally, eight lipids, including eicosenoic acid, eicosapentaenoic acid, PS(36:1), PS(36:4), PE(34:1), PC(34:2), PE(36:4), and PI(38:5), also showed contrasting change trends between the two DN models. These differences in metabolic phenotype should be considered when selecting animal models for drug development. For example, HFD/STZ-induced diabetic rats may be not appropriate for testing SGLT2-targeted drugs due to the low expression of the target in the renal medulla.

The present study has some limitations that should be noted. One of the main limitations is that the number size (n = 6) for each group is relatively small, which may impact the statistical power of the results and affect the robustness of the findings. To account for inter-animal variability, it would be ideal to have a larger sample size. Another limitation is that only male animals were used in this study, and it remains unclear whether these findings can be generalized to female animals or humans. Further research is needed to examine the metabolic changes in female DN animal models and to determine the translational relevance of these findings to human populations. Despite these limitations, this study provides valuable insights into the region-specific metabolic changes in different DN animal models and can guide future research on the pathophysiology and treatment of DN.

## 5. Conclusions

To summarize, this study compared the renal metabolic disturbances in an HFD/STZ-induced DN rat model and a db/db DN mouse model using spatially resolved metabolomics-based AFADESI-MSI. A wide range of discriminating metabolites were identified in the HFD/STZ-induced DN rats and the db/db DN mice by comparing their corresponding controls, respectively. The results showed that glycolysis, lipid metabolism, the TCA cycle, carnitine metabolism, choline metabolism, and purine metabolism were altered in a region-specific manner in the kidneys of both DN models. However, the regional levels of glucose, AMP, eicosenoic acid, eicosapentaenoic acid, PS(36:1), PS(36:4), PE(34:1), PE(36:4), PC(34:2), and PI(38:5) were changed in opposite directions, suggesting a significant difference in the metabolic phenotypes in the HFD/STZ-induced DN rat model and the db/db DN mouse model. This study provides more comprehensive and detailed information on the differences in the molecular pathological features in the two frequently used animal models of DN.

## Figures and Tables

**Figure 1 metabolites-13-00324-f001:**
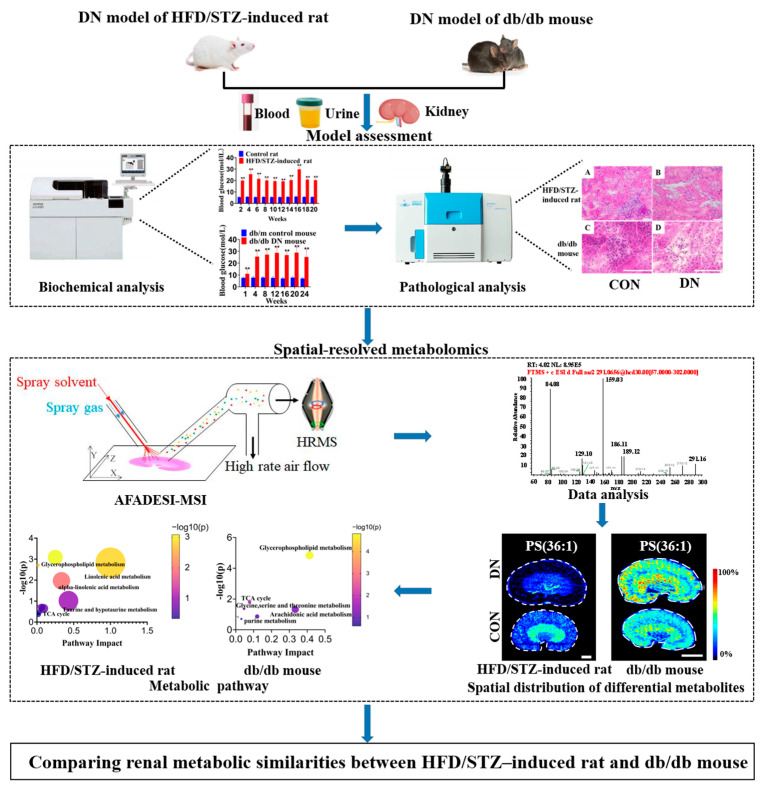
The research strategy for the comparison of region-specific metabolic changes in kidneys of HFD/STZ-induced diabetic rats and db/db mice. CON: control group; DN: diabetic nephropathy group; HFD/STZ-induced rat: high-fat diet feeding combined with intraperitoneal injection of low dose of STZ. Scale bar: 3 mm.

**Figure 2 metabolites-13-00324-f002:**
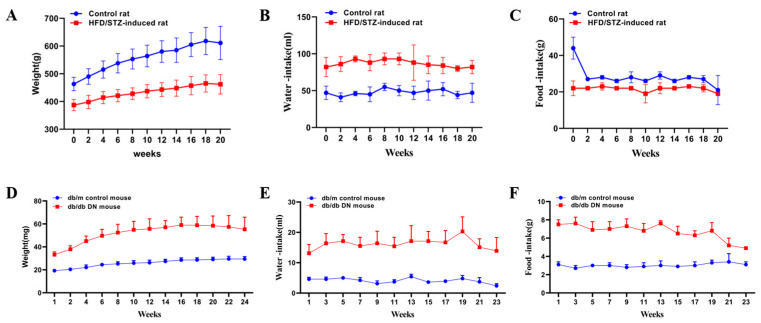
Body weight, water intake, and food intake from HFD/STZ-induced diabetic rats (**A**–**C**) and db/db mice (**D**–**F**). Control rat: control group rat;HFD/STZ-induced rat: high-fat diet feeding combined streptozotocin-induced diabetic nephropathy group rat; db/m control mouse: db/m control group mouse; db/db DN mouse: db/db diabetic nephropathy group mouse.

**Figure 3 metabolites-13-00324-f003:**
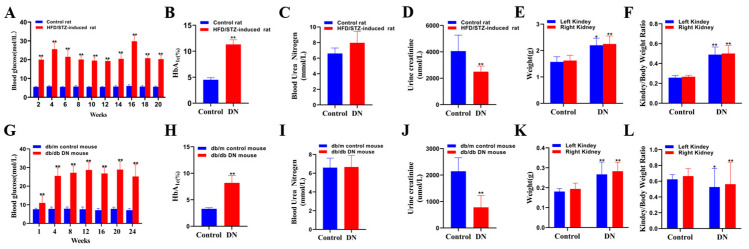
Physiological and biochemical indicators including blood glucose, glycosylated hemoglobin (HbA1c), urea nitrogen, urine creatinine, kidney weight, and kidney/body weight inHFD/STZ-induced rats (**A**–**F**) and db/db mice (**G**–**L**). Control rat: control group rat;HFD/STZ-induced rat: high-fat diet feeding combined streptozotocin-induced diabetic nephropathy group rat; db/m control mouse: db/m control group mouse; db/db DN mouse: db/db diabetic nephropathy group mouse. * *p* < 0.05, ** *p* < 0.01 compared with the control group.

**Figure 4 metabolites-13-00324-f004:**
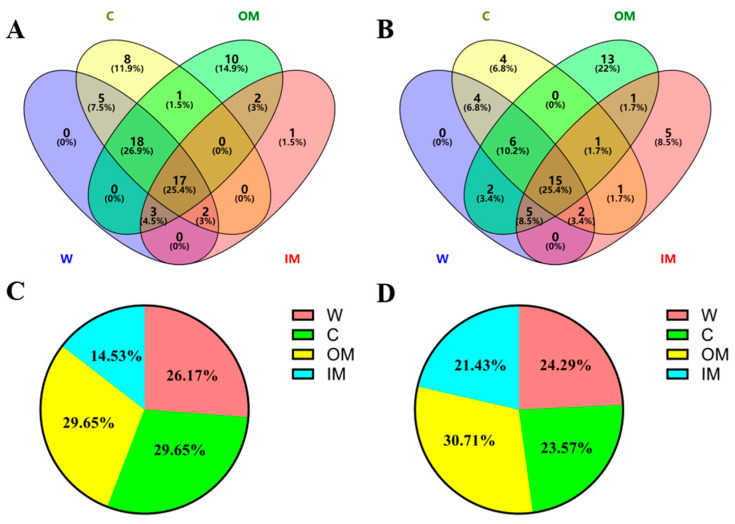
(**A**) Venn diagram of the number of discriminating metabolites detected in different regions of the kidneys in HFD/STZ-induced diabetic rats. (**B**) Venn diagram of the number of discriminating metabolites detected in different regions of the kidneys in db/db mice. (**C**) Percentage of discriminating metabolites identified in different regions of the kidneys in HFD/STZ-induced diabetic rats. (**D**) Percentage of discriminating metabolites found in different regions of the kidneys in db/db mice. W: whole; C: cortex; OM: outer medulla; IM: inner medulla.

**Figure 5 metabolites-13-00324-f005:**
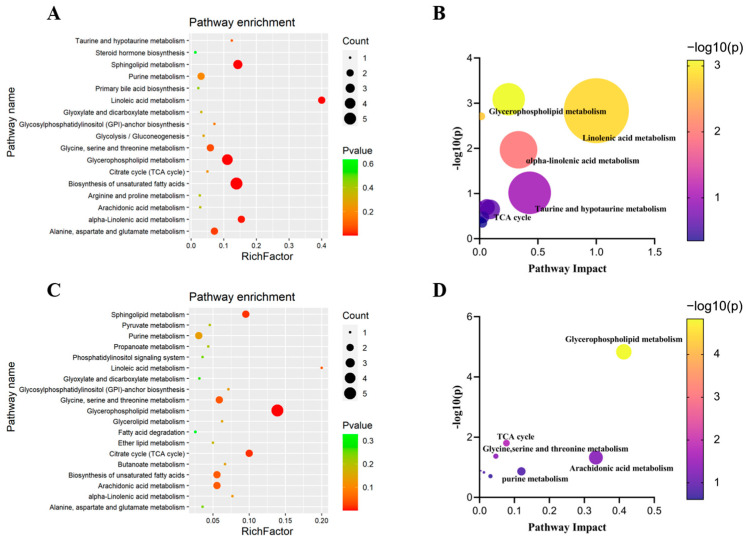
Pathway enrichment analysis of discriminating metabolites associated with diabetic nephropathy in HFD/STZ-induced diabetic rats (**A**,**B**) and db/db mice (**C**,**D**).

**Figure 6 metabolites-13-00324-f006:**
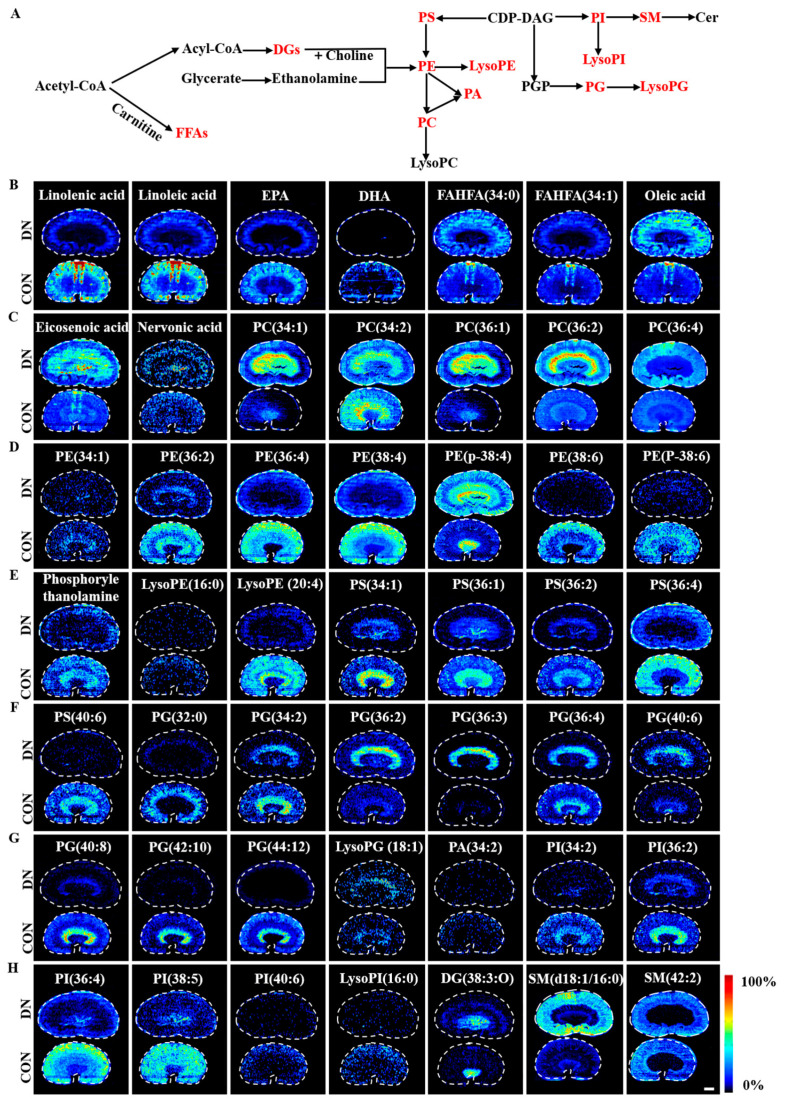
Simplified pathway diagram of lipid metabolism (**A**) and air flow-assisted desorption electrospray ionization–mass spectrometry images of metabolites involved in lipid metabolism in HFD/STZ-induced diabetic rats (**B**–**H**). CON: control group; DN: diabetic nephropathy groups; EPA: eicosapentaenoic acid; DHA: docosahexaenoic acid; FAHFAs: fatty acid esters of hydroxy fatty acids; PC: phosphatidylcholine; PE: phosphatidylethanolamine; PE(P-38:4): PE(P-18:0/20:4(5Z,8Z,11Z,14Z)); LysoPE: lysophosphatidylethanolamine; PS: phosphatidylserine; PG: phosphatidylglycerol; LysoPG: lysophosphatidylglycerol; PA: phosphatidic acid; PI: phosphatidylinositol; PI(36:2): PI(18:1(9Z):18:1(9Z)); LysoPI: lysophosphatidylinositol; DG: diacylglycerol; SM: sphingomyelin. Scale bar: 3 mm.

**Figure 7 metabolites-13-00324-f007:**
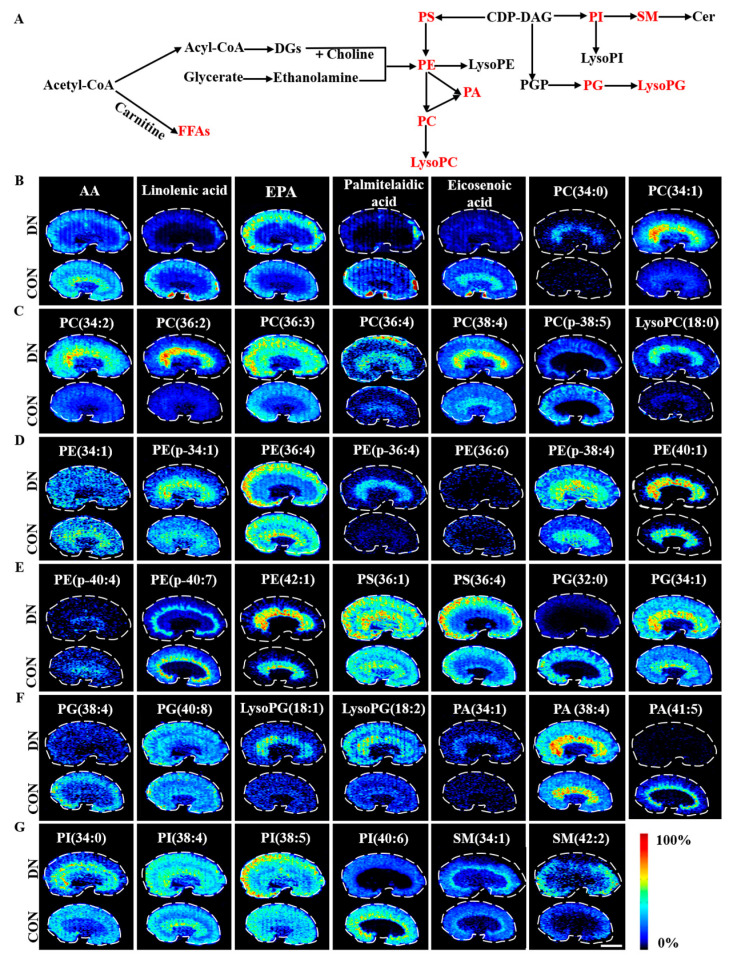
Simplified pathway diagram of lipid metabolism (**A**), and air flow-assisted desorption electrospray ionization–mass spectrometry images of metabolites involved in lipid metabolism in db/db mice (**B**–**G**). CON: control group; DN: diabetic nephropathy groups; AA: arachidonic acid; EPA: eicosapentaenoic acid; PC: phosphatidylcholine; LysoPC: lysophosphatidylcholine; PE: phosphatidylethanolamine; PS: phosphatidylserine; PG: phosphatidylglycerol; LysoPG: lysophosphatidylglycerol; PA: phosphatidic acid; PA(34:1): 16:0/18:1(9Z)); PI: phosphatidylinositol; SM: sphingomyelin; SM (34:1): SM(d18:1/16:0); SM(42:2): SM(d18:1/24:1(15Z)). Scale bar: 3 mm.

**Figure 8 metabolites-13-00324-f008:**
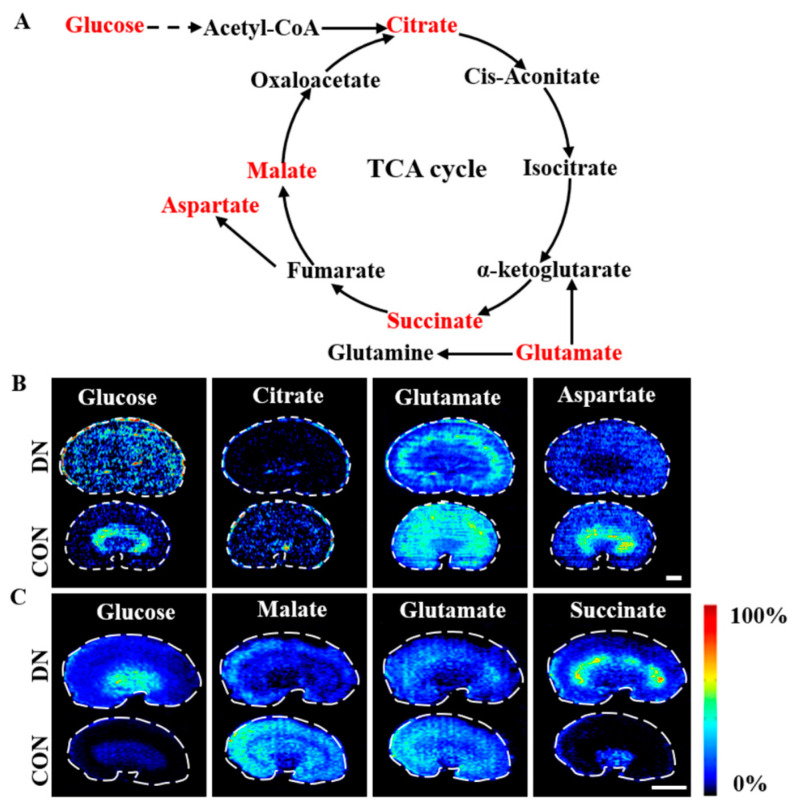
Simplified pathway diagram of glycolysis and TCA cycle (**A**) and air flow-assisted desorption electrospray ionization–mass spectrometry images of metabolites involved in glycolysis and TCA cycle in HFD/STZ-induced diabetic rats (**B**) and db/db mice (**C**). CON: control group; DN: diabetic nephropathy groups. Scale bar: 3 mm.

**Figure 9 metabolites-13-00324-f009:**
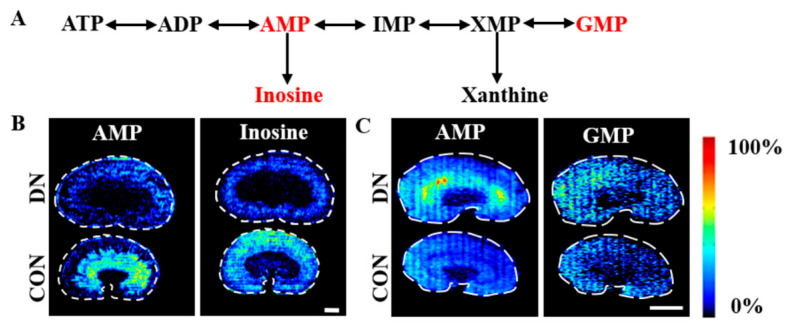
Simplified pathway diagram of purine metabolism (**A**) and air flow-assisted desorption electrospray ionization–mass spectrometry images of metabolites involved in purine metabolism in HFD/STZ-induced diabetic rats (**B**) and db/db mice (**C**). CON: control group; DN: diabetic nephropathy groups; ATP: adenosine triphosphate; ADP: adenosine diphosphate; AMP: adenosine monophosphate; IMP: inosine monophosphate; XMP: xanthosine monophosphate; GMP: guanine monophosphate. Scale bar: 3 mm.

**Figure 10 metabolites-13-00324-f010:**
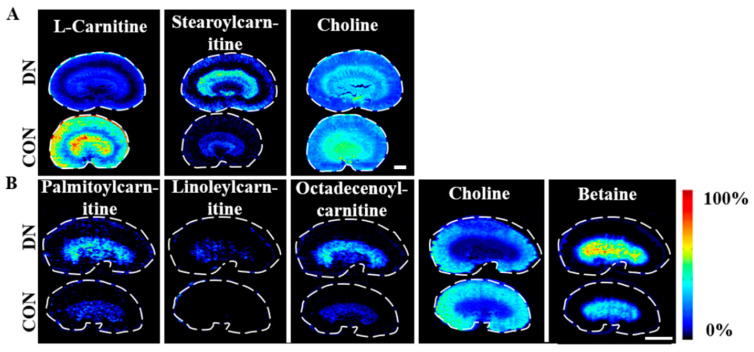
Air flow-assisted desorption electrospray ionization–mass spectrometry images of metabolites involved in carnitine and choline metabolism in HFD/STZ-induced diabetic rats (**A**) and db/db mice (**B**). CON: control group; DN: diabetic nephropathy groups. Scale bar: 3 mm.

**Figure 11 metabolites-13-00324-f011:**
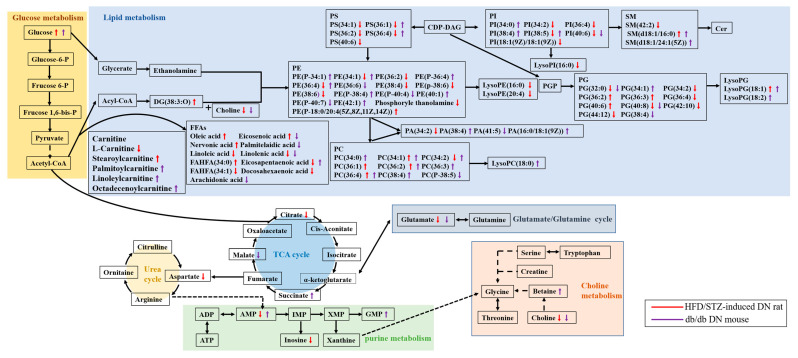
Metabolites and metabolic pathways associated with DN in HFD/STZ-induced diabetic rats and db/db mice. DN: diabetic nephropathy group. PC: phosphatidylcholine; PE: phosphatidylethanolamine; PS: phosphatidylserine; PI: phosphatidylinositol; PG: phosphatidylglycerol; DG: diacylglycerol; SM: sphingomyelin; LysoPG: lysophosphatidylglycerol; LysoPI: lysophosphatidylinositol; PA: phosphatidic acid; FAHFAs: fatty acid esters of hydroxy fatty acids. ATP: adenosine triphosphate; ADP: adenosine diphosphate; AMP: adenosine monophosphate; IMP: inosine monophosphate; XMP: xanthosine monophosphate; GMP: guanine monophosphate. ⬆: Up-regulated; ⬇: Down-regulated.

## Data Availability

Data are contained within the supplementary material.

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
