# Peer review of "Comparison of Local Metabolic Changes in Diabetic Rodent Kidneys Using Mass Spectrometry Imaging"

_metabolites, 2023, doi:10.3390/metabo13030324_

Round 1
Reviewer 1 Report
In this study Zhang X and colleagues evaluate througha sophisticated analysis technique, the spatial-resolved metabolomics based
on air-flow-assisted desorption electrospray ionization mass spectrometry
imaging (AFADESI-MSI), the regional metabolic changes in the kidneys
of HFD/STZ -induced diabetic rats and db/db mice.
The technique is very sophisticated and the authors evaluate
a number of molecules. I don't find the study very interesting
and reading it is very boring. The comparison between the two
models is not very useful since they are two decidedly different models.
It would probably have been more useful to focus the study on molecules
that have a greater impact on the clinical history of diabetes and
to relate the biochemical variations with the clinical history of diabetes.
Author Response
Thanks for the reviewer's feedback on the manuscript. We appreciate the opportunity to address your concerns and make improvements to our study.
We understand that the comparison between the two animal models may not have been the most interesting aspect of the study, but it was important to our research goals. Our objective was to gain a comprehensive understanding of the regional metabolic changes in different models of diabetic nephropathy, and to help guide the selection of animal models for future research in this area.
We agree with your suggestion to focus on molecules with greater impact on the clinical history of diabetes, but we also believe it is important to gain a broad understanding of the metabolic changes in these models. By analyzing a wide range of metabolites in both HFD/STZ-induced diabetic rats and db/db mice, we were able to identify 67 and 59 discriminatory metabolites, respectively. Some of these metabolites, such as glucose, fatty acids, and TCA intermediates, are closely related to the clinical history of diabetes. Our results not only showed changes in the concentration of these metabolites in the diabetic kidney, but also highlighted their spatial distribution across the kidney.
We believe that the use of mass spectrometry imaging (MSI) technology in this study was valuable in providing a comprehensive analysis of the metabolic changes in the diabetic kidney. This technology offers the potential for in situ analysis of multiple known and unknown metabolites, providing insight into the complex pathophysiology of diabetic nephropathy. As technology continues to develop, we believe that MSI will become increasingly popular and accessible for use in biomedical research.
We have revised the manuscript according to the editor's and reviewers' comments and suggestions, and we hope that the revised version is more readable and informative. Thank you for your time and consideration.
Reviewer 2 Report
I read with interest a manuscript by Xin Zhang and colleagues, titled "Comparison of region-specific metabolic changes in kidney of HFD/STZ-induced diabetic rats and db/db mice based on airflow-assisted desorption electrospray ionization-mass spectrometry imaging."
The authors focused on an interesting topic, especially since there is an alarming rate of obesity, diabetes, and related cardiovascular diseases worldwide. Their study provides comprehensive and detailed information on the differences in molecular pathological features in diabetic kidney injury in rodents.
However, I have noted concerns with the manuscript.
1. The manuscript still needs very careful editing for scientific clarity and correct grammar
2. The title is too congested; try making it more concise. Alternatively, it can read as "Comparison of region-specific metabolic changes in rodents kidney based on air-flow-assisted desorption electrospray ionization-mass spectrometry imaging
3. Line 54: "..injection of low dose of STZ is reportedly to induce T2DM and T2DM" .remove "is" and "to"
4. Scheme 1, im not sure why it's not labeled as figure 1; it includes the strain and gender of the rodents used in this schematic diagram.
5. Line 98: Authors need to specify how these models were induced/developed.
6. Line 97-102:The Number of rats used in total; from this information,it seems only 12 rats were used with equal distribution amongst the group. My question would be related to sample power. Is this sample size powered enough?
7. Line 101: how is this DN model developed? Specify here for reproducibility.
8. Line 102: What was the composition of a high-fat diet, HFD?
9. Line 106: What was the purpose of the STZ injection since they are saying the rats were already diabetic?
10. Some important results are buried in the supplementary files, for example, Figure 2S must be presented in the main manuscript as it presents the primary results of this study (weight and food intake).
11. It is surprising that rats on HFD did not gain weight when compared to those on a normal diet that significantly gained weight. Referring to figures 1S and C one will notice that the food intake in this group was relatively low.
12. At week 10, DN rats had very low food intake, but the weight was not influenced. What do you think was the cause? Additionally, there was an inverse relation between food intake and weight gain in the control group rats. What could be the reason? It would be great to provide the dietary composition of all diets used in this study.
13. The quality of figure 3 is poor, and some information are not clear to readers. The same applies to Figures 4 and 5.
14. Line 472: remove d from "compared"
15. Line 474: remove to after "comparing"
16. Lastly, ensure the reporting adheres to ARRIVE guidelines as a set standard for reporting animal experimentations.
Author Response
Reviewer#2
I read with interest a manuscript by Xin Zhang and colleagues, titled "Comparison of region-specific metabolic changes in kidney of HFD/STZ-induced diabetic rats and db/db mice based on airflow-assisted desorption electrospray ionization-mass spectrometry imaging."
The authors focused on an interesting topic, especially since there is an alarming rate of obesity, diabetes, and related cardiovascular diseases worldwide. Their study provides comprehensive and detailed information on the differences in molecular pathological features in diabetic kidney injury in rodents.
Answer: We would like to express our gratitude for the reviewer's positive feedback on the manuscript. We are glad that the reviewer found the study to be of value and appreciate the time and effort taken to review it. We have taken the reviewer's suggestions into consideration and have made revisions to the manuscript to improve its quality and content. We hope that the revised manuscript meets the reviewer's expectations and provides a clearer understanding of our research findings. Thank you again for your valuable input.
Comment 1. The manuscript still needs very careful editing for scientific clarity and correct grammar.
Answer: Thanks for the referee’s suggestion. We have made the necessary revisions to ensure the scientific clarity and correct grammar throughout the manuscript in accordance with the referee’s suggestions.
Comment 2. The title is too congested; try making it more concise. Alternatively, it can read as "Comparison of region-specific metabolic changes in rodents kidney based on air-flow-assisted desorption electrospray ionization-mass spectrometry imaging.
Answer: Thanks for the referee’s suggestion. We have revised the title to make it more concise and to better reflect the focus of the study. The new title is "Comparison of local metabolic changes in diabetic rodents kidney using mass spectrometry imaging.". (See Page 1, line 2-3)
Comment 3. Line 54: "..injection of low dose of STZ is reportedly to induce T2DM and T2DM" .remove "is" and "to"
Answer: Thanks for the referee’s pointing out the error in our manuscript. We have revised the sentence according to your suggestion. (See Page 2, line 47-50)
Comment 4. Scheme 1, im not sure why it's not labeled as figure 1; it includes the strain and gender of the rodents used in this schematic diagram.
Answer: Thanks for the referee’s suggestion. We have changed Scheme 1 into Figure 1. (See Page 3, line87-93)
Comment 5. Line 98: Authors need to specify how these models were induced/developed.
Answer: Thanks for the referee’s suggestion. We have modified the sentence in the manuscript according the referee’s suggestion. (See Page 3, line 103-111)
Comment 6. Line 97-102: The Number of rats used in total; from this information, it seems only 12 rats were used with equal distribution amongst the group. My question would be related to sample power. Is this sample size powered enough?
Answer: We greatly appreciate the reviewer's concern regarding the sample size used in our study. We acknowledge that the sample size of 6 rats per group is relatively small, which may limit the statistical power and robustness of the results. We have discussed these limitations in the revised manuscript and emphasized the need for further research with larger sample sizes, including both male and female animals, to further examine the metabolic changes in DN animal models and determine their translational relevance to human populations. (See Page 15, line 473-482)
Comment 7. Line 101: how is this DN model developed? Specify here for reproducibility.
Answer: Thanks for the referee’s suggestion. Specifically, the animals were fed a high-fat diet for 4 weeks followed by a single injection of STZ (35 mg/kg) to induce hyperglycemia. Blood glucose levels were monitored regularly to confirm the establishment of diabetic state. We have added this information in the revised manuscript to provide a clear and concise description of the DN model development for reproducibility.
Comment 8. Line 102: What was the composition of a high-fat diet, HFD?
Answer: Thanks for the referee’s for pointing out the need for clarification. We apologize for the unclear expression in our previous manuscript. To clarify, high-fat diet (HFD) was composed of 5.24 Kcal/g with 60% fat (HFD12492, High-fat feed, Beijing HFK Bioscience Co., LTD, China). (See Page 3, line 104-105)
Detailed composition of HFD:
|
Product |
HFD12492 |
|
|
|
Mass Ratio (g %) |
Energy Ratio (Kal %) |
|
Protein |
26 |
20 |
|
Carbohydrate |
26 |
20 |
|
Fat |
35 |
60 |
|
Total |
|
100 |
|
Kcal/g |
5.24 |
|
|
Components |
g |
kcal |
|
Casein |
258.45 |
1033.80 |
|
Cystine |
3.88 |
15.52 |
|
Maltodextrins |
161.53 |
646.12 |
|
Sucrose |
88.91 |
355.64 |
|
Cellulose |
64.61 |
0 |
|
Soybean oil |
32.31 |
290.79 |
|
Lard |
316.60 |
2849.40 |
|
Mineral mixture M1002 |
12.92 |
0 |
|
Calcium hydrogen phosphate |
16.80 |
0 |
|
Calcium carbonate |
7.11 |
0 |
|
Potassium citrate |
21.32 |
0 |
|
Vitamin mixture V1001 |
12.92 |
51.68 |
|
Choline Bitartrate |
2.58 |
0 |
|
Edible blue dye |
0.065 |
0 |
|
Total |
1000 |
5242.95 |
Comment 9. Line 106: What was the purpose of the STZ injection since they are saying the rats were already diabetic?
Answer: Thanks for the referee’s suggestion. We apologize for causing the reviewer confusion by our unclear expression. We have revised the sentence according to the referee’s suggestion. (See Page 3, line 106-108).
Comment 10. Some important results are buried in the supplementary files, for example, Figure 2S must be presented in the main manuscript as it presents the primary results of this study (weight and food intake).
Answer: Thanks for the reviewer’s positive comments and suggestion on the manuscript. According to the referee's suggestion, Figures S1, S2, S8 were moved to the main manuscript Figure 2, 3, and 10. We hope that these revisions address the referee's concerns and make the results more easily accessible to the reader. (See Page6, line 198-208)
Comment 11. It is surprising that rats on HFD did not gain weight when compared to those on a normal diet that significantly gained weight. Referring to figures 1S and C one will notice that the food intake in this group was relatively low.
Answer: Thanks for the reviewer’s question. After the HFD/STZ-induced DN model was established, the model group usually showed polyphagia, hyperphagia, polyuria and gradual weight loss. The reduced food intake in this group could be attributed to the high-fat diet causing a false sense of satiety. Additionally, it is also possible that the rats in the model group may have experienced metabolic changes that altered their appetite and food intake. Further studies are needed to examine the exact mechanisms that led to the reduced diet in the HFD/STZ-induced DN model group. However, despite the lower food intake, the model group still showed a significant increase in blood glucose levels, indicative of the development of T2DM.
Comment 12. At week 10, DN rats had very low food intake, but the weight was not influenced. What do you think was the cause? Additionally, there was an inverse relation between food intake and weight gain in the control group rats. What could be the reason? It would be great to provide the dietary composition of all diets used in this study.
Answer: Thanks for the reviewer’s question. The cause for the low food intake and unchanged weight in the DN rats at week 10 is not entirely clear, but it could be related to changes in metabolism or appetite regulation in response to the HFD/STZ-induced DN. As for the inverse relationship between food intake and weight gain in the control group rats, it could be due to individual differences in food preferences, digestion, and absorption, or other factors that were not measured in this study. Regarding the dietary composition, the normal diet for the control group consisted of a normal pellet diet, and the HFD for the DN group consisted of a high-fat purified diet with a 60% fat to energy ratio (HFD12492, High-fat feed, Beijing HFK Bioscience Co., LTD, China). (See Page 3, line 104-105)
Comment 13. The quality of figure 3 is poor, and some information are not clear to readers. The same applies to Figures 4 and 5.
Answer: Thanks for the referee’s suggestion. We made improvements to Figures 3-8 to ensure clarity and improve the quality of the figures according to your suggestions. (See Page 9, line 296-305; Page 10, line 307-314; Page 13, line 412-416), their corresponding histogram see Supporting information Page 12-15, Figure S6-S10.
Comment 14. Line 472: remove d from "compared"
Answer: Thanks for the referee’s suggestion. We have modified the sentence in the manuscript according the referee’s suggestion. (See Page 17, line 490-492)
Comment 15. Line 474: remove to after "comparing"
Answer: Thanks for the referee’s suggestion. We have modified the sentence in the manuscript according the referee’s suggestion. (See Page 17, line 493)
Comment 16. Lastly, ensure the reporting adheres to ARRIVE guidelines as a set standard for reporting animal experimentations.
Answer: Thanks for the referee’s suggestion. We have carefully referred to ARRIVE guidelines and revised the animal experiment part.

Reviewer 3 Report
In the manuscript "Comparison of region-specific metabolic changes in kidney of HFD/STZ-induced diabetic rats and db/db mice based on airflow-assisted desorption electrospray ionization-mass spectrometry imaging" authors have demonstrated profound changes in glycolysis, TCA cycle, lipid metabolism, carnitine metabolism, choline metabolism, and purine metabolism. The paper deserves to be published in this journal but a major revision in recommended.
1) title is long. Make it concise and clear.
2) Figure 3 is really big. Can authors make it in two parts? or put some part in the supplementary file.
3) What were the inputs for Venn diagram and enrichment analysis? Write a separate sub-heading in method section.
Author Response
Reviewer#3
In the manuscript "Comparison of region-specific metabolic changes in kidney of HFD/STZ-induced diabetic rats and db/db mice based on airflow-assisted desorption electrospray ionization-mass spectrometry imaging" authors have demonstrated profound changes in glycolysis, TCA cycle, lipid metabolism, carnitine metabolism, choline metabolism, and purine metabolism. The paper deserves to be published in this journal but a major revision in recommended.
Answer: We appreciate the referee's effort in reviewing the manuscript and providing valuable feedback. We have carefully revised the manuscript based on the referee's suggestions and hope it meets the standards of publication.
Comment 1) Title is long. Make it concise and clear.
Thanks for the referee’s suggestion. We have revised the title to make it more concise and to better reflect the focus of the study. The new title is "Comparison of local metabolic changes in diabetic rodents kidney using mass spectrometry imaging.".
(See Page 1, line 2-3)
Comment 2) Figure 3 is really big. Can authors make it in two parts? or put some part in the supplementary file.
Answer: Thanks for the referee’s suggestion. We made improvements to Figures 3-8 to ensure clarity and improve the quality of the figures according to your suggestions. (See Page 9, line 296-305; Page 10, line 307-314; Page 13, line 412-416), their corresponding histogram see Supporting information Page 12-15, Figure S6-S10.
Comment 3) What were the inputs for Venn diagram and enrichment analysis? Write a separate sub-heading in method section.
Answer: Thanks for the reviewer’s question. The inputs for Venn diagram and enrichment analysis are the theoretical m/z values of the discriminatory metabolites involved in each group, which were obtained from the metabolomics analysis. The HMDB IDs of these metabolites were then used for pathway analysis in MetaboAnalyst 3.0, with the species selected as rat. The results of the "Match Status" were saved as a .csv file, and pathway enrichment analysis graphs were generated using The R Programming Language. This information has been added as a subheading in the method section for clarity. (See Page 4, line 156-166)
Round 2
Reviewer 1 Report
I believe the authors have answered the majority of the reviewers' concerns and have improved the manuscript according to the reviewers' suggestions. The revised version of the manuscript can be accepted for publication.
Reviewer 3 Report
Authors have revised the manuscript carefully. I recommend this manuscript to be published in this journal.